# A Driver Never Works Alone—Interplay Networks of Mutant p53, MYC, RAS, and Other Universal Oncogenic Drivers in Human Cancer

**DOI:** 10.3390/cancers12061532

**Published:** 2020-06-11

**Authors:** Maria Grzes, Magdalena Oron, Zuzanna Staszczak, Akanksha Jaiswar, Magdalena Nowak-Niezgoda, Dawid Walerych

**Affiliations:** 1Mossakowski Medical Research Centre Polish Academy of Sciences A. Pawinskiego 5, 02-106 Warsaw, Poland; mgrzes@imdik.pan.pl (M.G.); moron@imdik.pan.pl (M.O.); zstaszczak@imdik.pan.pl (Z.S.); ajaiswar@imdik.pan.pl (A.J.); 2Central Clinical Hospital of the Ministry of the Interior and Administration, Woloska 137, 02-507 Warsaw, Poland; magnowak@gmail.com

**Keywords:** cancer, oncogene, transformation, mutant p53, MYC, RAS

## Abstract

The knowledge accumulating on the occurrence and mechanisms of the activation of oncogenes in human neoplasia necessitates an increasingly detailed understanding of their systemic interactions. None of the known oncogenic drivers work in isolation from the other oncogenic pathways. The cooperation between these pathways is an indispensable element of a multistep carcinogenesis, which apart from inactivation of tumor suppressors, always includes the activation of two or more proto-oncogenes. In this review we focus on representative examples of the interaction of major oncogenic drivers with one another. The drivers are selected according to the following criteria: (1) the highest frequency of known activation in human neoplasia (by mutations or otherwise), (2) activation in a wide range of neoplasia types (universality) and (3) as a part of a distinguishable pathway, (4) being a known cause of phenotypic addiction of neoplastic cells and thus a promising therapeutic target. Each of these universal oncogenic factors—mutant p53, KRAS and CMYC proteins, telomerase ribonucleoprotein, proteasome machinery, HSP molecular chaperones, NF-κB and WNT pathways, AP-1 and YAP/TAZ transcription factors and non-coding RNAs—has a vast network of molecular interrelations and common partners. Understanding this network allows for the hunt for novel therapeutic targets and protocols to counteract drug resistance in a clinical neoplasia treatment.

## 1. Oncogene Cooperation in a Game of Multistep Carcinogenesis 

Since the discovery of the first cellular oncogenes in 1970s the idea that certain genes may promote cancer more efficiently than others has been sustained. Nowadays, the cohorts of thousands of patients in initiatives such as TCGA and ICGC (mostly untreated primary neoplasias) [1,2] or IMPACT (advanced and metastatic neoplasias) [3] result in lists of dozens of frequently mutated loci, present in various transformed tissues. Yet, questions that date back to the initial discoveries remain at least partially unanswered: Which of the neoplasia-derived changes are decisive for its development and metastasis? Does a more frequent change mean it is more important? What network of oncogenic pathways is an essential setup for the initiation and driving of neoplasia at a given stage? We elaborate on these questions not only by focusing on the properties of individual oncogenes but rather by discussing the most representative examples of molecular interplay between universal and frequently studied oncogenic drivers.

As the most universally activated and popular research subjects we considered tumor protein 53 *(TP53),* rat sarcoma viral oncogene homolog (*RAS)* and avian myelocytomatosis virus oncogene homolog (*MYC)* family genes and their molecular networks. Overall, *TP53* is the most frequently mutated gene in human neoplasias (42–43% mutations [1,3]) with well-established tumor-suppressive activities, while the hot-spot missense mutant variants have been shown to possess oncogenic gain-of-function properties (Figure 1). It is a unique setup of a tumor suppressor and a proto-oncogene “in one”, which remains somewhat controversial despite hundreds of studies covering the subject. This is perhaps in part due to a lack of experiments clearly demonstrating the mutant p53 individual, de novo transforming ability in human normal cell multistep carcinogenesis in vitro setups (see the section about *TP53* mutants). However, as the Kirsten rat sarcoma viral oncogene homolog (*KRAS*) gene example demonstrates, even the gene, which is known as a powerful and indispensable component of both classic and modern multistep carcinogenesis models [4,5], is dependent on particular partners and molecular events to be effective in de novo cell transformation [6,7]. The main *MYC* family protooncogene, cellular avian myelocytomatosis virus oncogene homolog (*CMYC*), demonstrates yet another aspect of molecular tumorigenesis. It controls a vast transcriptional program that is important for normal cells, and even without overexpression it is regarded as one of the most addictive oncogenes in human tumorigenesis [8,9] (Figure 2), demonstrating its independent de novo tumorigenicity in human cells in vitro [7,10]. The cases of the other driver proteins and RNAs which we describe in the following paragraphs are similar—none of which can individually be regarded as definitive for a cellular transformation and metastasis. 

This aspect is dramatically reflected in the anti-cancer therapeutics field. In the past decades, the identification of the major molecular hubs responsible for cellular transformation was supposed to allow therapies targeting these hubs to effectively suppress neoplasia. The reality is that over 30 years since discoveries of *TP53, KRAS* and *CMYC* there is no single standard therapy in the clinics directly targeting any of these genes and proteins. In the case of the most successful anti-oncogene therapies—such as inhibitors of the proteasome machinery, v-raf murine sarcoma viral oncogene homolog B (BRAF) or phosphoinositide 3 (PI3) kinase networks, or cell membrane receptors—the success is partial, as the therapies are highly specific to neoplasia types and are prone to compensatory resistance mechanisms (see the oncogene-specific paragraphs for details and citations). Understanding inter-oncogenic signaling and inter-cellular relationships in a neoplastic tissue will lead to progress in defining new therapeutic protocols and counteracting drug resistance. 

## 2. Mutant p53 Gain-of-Function 

Wild-type (WT) p53 is a tumor suppressor protein, which upon induction by stress factors acts mainly as a teterameric transcription factor which induces multiple events—such as metabolic adaptation, proliferation arrest or apoptosis. Due to these p53 properties mutations in the *TP53* gene in the context of cancer may lead to three events: loss of function (LOF) of WT p53 tumor suppressor protein which mainly involves loss of transcription factor activity, and—especially in the case of prevalent missense mutations—dominant-negative effect over WT p53 (DN) in a tetrameric protein and gain-of-function (GOF) which can actively drive neoplasia [11,12]. The details of this last mechanism still spark controversies, although it was directly proven nearly three decades ago [13], followed by key mouse model experiments [14,15] and a spectrum of mechanistic studies [16]. Possible causes of the controversies are: hotspot, missense *TP53* GOF mutants were—perhaps only by negligence—not shown to be decisive components of the multistep carcinogenesis in human normal cells models in vitro (only truncated p53 DN mutants or WT p53 inhibition were used) [4,6,7,17,18,19] and there are specific neoplasia models which, for reasons yet unknown, do not show GOF of missense *TP53* mutants [20,21]. Nevertheless, the vast majority of current knowledge points to mutant p53 GOF mechanisms as pivotal to oncogenic involvement of p53 in key pro-neoplastic molecular pathways (Figure 1).

As studies both in vitro and in vivo have demonstrated, to efficiently exert GOF, p53 mutants have to be accumulated in cells [22,23]. One of the major stabilizing factors for p53 mutants is a complex of molecular chaperones HSP90 and HSP70 [22,24,25]. They lead to impairment of an access of MDM2 and CHIP ubiquitin ligases to p53, and as the consequence—to the mutant p53 stabilization [22,26]. p53 interactions with other oncogenic proteins are critical to executing the GOF activity. Such is the case of the mentioned chaperone machinery or PIN1 peptidyl-prolyl isomerase which support both WT and mutant p53 [25,27]. However, the main involvement of mutant p53 in intercepting other oncogenic pathways is via its specific interactions, possible due to mutation-related structural changes in a globular, Zn ion-stabilized, p53 DNA binding domain. Apart from a direct inhibition of tumor suppressors, such as p63 and p73 [28], this mechanism mainly leads to aberrant augmentation of transcription factor activity. One of them is CMYC, found to specifically interact with R249S mutant p53 phosphorylated at the introduced serine, which attracts PIN1 binding, and leads to enhancement of the CMYC-driven transactivation [29]. Mutant p53 interacts with the NFY family of transcription factors and allows p300 acetyltransferase to be recruited to NFY gene promoters [30,31], which may also involve recruitment of YAP transcription factor [32]. CMYC was found as one of the mutant p53-NFY axis targets [33], in concordance with early reports [34]. Several studies in different experimental models showed an interaction of mutant p53 with ETS2 transcription factor which leads to upregulation of pro-neoplastic ETS2 target genes [35,36]. Mutant p53 binds SREBP transcription factors and augments their activity, which promotes cancer phenotypes via a mevalonate pathway [37]. Mutant p53 intercepts NRF2 (NFE2L2) protein housekeeping activity and antioxidant response in a selective manner and pushes NRF2 onto a tumor-promoting track [38,39]. Mutant p53 is also known to cooperate with other oncogenic transcription factors: NFkB [40], STAT3 [41], AP-1 or β-catenin (mentioned further in the dedicated paragraphs below), with more examples reviewed elsewhere [16,42]. 

Outside of the transcriptional complexes mutant p53 was also found to exert other, wide influences in neoplastic cells. It was shown to interfere with DNA-repair mechanisms driven by MRE11, leading to genomic instability [43], to interfere directly or indirectly with the microRNA maturation complex to alter the landscape of micro RNA expression [38,44] or to be involved in inflammatory and unfolded protein responses due to its cytoplasmic interactions [45,46]. Known cooperation of mutant p53 with RAS family oncoproteins is more indirect—both pathways contribute to progression of neoplasia in parallel, e.g., in pancreatic cancer models [47,48]. As in the case of most tumor suppressors, the indirect interplay of p53 with CMYC, KRAS or telomerase also includes a passive LOF contribution—when mutated or otherwise inactivated WT *TP53*, which normally may inhibit the oncogenes, loses this function and thus allows the oncogenes to promote neoplastic phenotypes [49,50,51].

The only dedicated agent in clinical trials against mutant p53 is APR-246 (PRIMA-1MET) which restores features of WT p53 activity, but also inhibits mutant p53 GOF [52]. The research so far suggests that the efficiency of APR-246 treatment is increased by chemotherapy or compounds targeting oncogenic partners of mutant p53 [52]. The suggestions for testing include inhibitors of chaperones, co-chaperones or cooperating histone deactylases (HDACs) to impair upstream activation and stabilization of mutant p53 [53,54] as well as co-targeting mutant p53 and the pathways controlled by mutant p53—such as epigenetic modification inhibitors [36], proteasome inhibitors [38], or statins—targeting the mevalonate pathway [37]. 

## 3. CMYC and MYC Family of Proteins

One of the cancer-involved proteins with the longest research history of nearly 40 years is CMYC [55]. CMYC belongs to the MYC protein family commonly deregulated in human neoplasia - comprising also NMYC and LMYC [8]. CMYC is a basic helix-loop-helix leucine zipper (bHLH-LZ) transcription factor that heterodimerizes with MAX to bind DNA and holds a strategic position in growth-promoting pathways involved in vital cellular processes [8] (Figure 2). In adult tissues MYC proteins are present in low level, as the MYC expression is tightly regulated [56]. Oncogenic activation of CMYC may occur through direct mechanisms such as mutations, gene amplifications, chromosomal translocations or enhancer insertions [9]. Point mutations in *CMYC* are relatively rare, and their role in driving tumorigenesis is questionable, however the correlation of T58 and F138 mutations with worse survival outcome in B-cell lymphoma patients was reported [57]. CMYC alterations are usually exclusive with BRAF, APC, PTEN and PIK3CA alterations [58]. Expression of other members of the *MYC* family, *NMYC* and *LMYC,* was observed in neuroblastoma and lung cancer, and subsequently in a number of other cancer types [9].

An important means of CMYC pathway interception by other oncogenic factors is CMYC upregulation by upstream growth-regulatory or signaling pathways [8]. Firstly, signaling pathways may act as inducers (Hedgehog, WNT pathway alone or in combination with NOTCH and JAK-STAT3) or repressors (as TGFβ) of the *CMYC* transcription [59,60]. The second possibility is increasing the efficiency of *CMYC* translation, which can be triggered by MAPK-HNRPK and mTORC1-S6K1 and inhibited by the MAPK-FOXO3A signaling cascade [61,62,63]. Finally, the cooperation of PI3K and RAS pathways affects the stability of CMYC throughout the posttranslational phosphorylation cascade required for CMYC release from DNA and degradation by the proteasome [64].

In cancer, the process of aberrant RAS signaling together with PIN1 activation and PP2A suppression leads to the accumulation of active CMYC [65]. Koh et al. reported a feed-forward loop between CMYC and hTERT (telomerase) in CMYC-driven lymphoma, in which the *hTERT* transcription was upregulated by CMYC. In turn hTERT stabilized CMYC protein to promote lymphomagenesis [66]. Several long non-coding RNAs (lncRNAs) contribute to CMYC stability and expression control—for example LINC-ROR by binding to AUF1 increases the CMYC stability, whereas overexpression of GAS5 inhibits the CMYC translation (for more examples see the non-coding RNAs paragraph) [67]. CMYC also creates an interesting regulatory network with micro RNAs (miRNAs). Chang et al. showed that CMYC induction results in a prevalent repression of miRNA expression [68]. Restoring their expression inhibited lymphoma cells growth in vivo, suggesting their importance in the CMYC tumorigenic program. In turn, the activation of transcription factor YAP deregulates miRNAs inhibiting CMYC, leading to the elevated *CMYC* expression [69].

*CMYC* overexpression is considered to be a molecular hallmark of many cancer types, indicating poor clinical outcome and contributing to tumor initiation and progression [70]. Numerous proliferative activities driven by CMYC include transcriptional cyclins upregulation, cell cycle suppressor downregulation, activation of anaerobic metabolism genes, BCL family apoptosis inhibitors or RNA-related processes—including oncogenic component of the cellular splicing machinery—PRMT5 [71,72,73,74,75]. Notably, the CMYC overexpression alone was insufficient to induce a neoplastic transformation and in normal cells led to apoptosis or proliferative arrest [76]. Other parallel oncogenic events are needed: loss of WT p53 or p19ARF as well as overexpression of BCL-2 and appropriate epigenetic context [70]. The studies conducted in CMYC-driven tumor cells and tumor nodules by Kress et al. and Walz et al. enabled distinguishing the specific sets of genes directly deregulated by CMYC from genes deregulated via secondary mechanisms [77,78]. 

The conclusion from numerous studies is that tumors become CMYC-addicted and driven even if CMYC is not the initiating oncogene [70]. The efforts to directly target CMYC have proven challenging because the protein structure is difficult to target with small molecules and its vast role in normal tissue homeostasis is a source of toxicity. Thus, alternative approaches, such as targeting of the *CMYC* transcription and translation, as well as targeting CMYC stability and the CMYC-MAX complex or upstream oncogenic network are being developed [79,80].

## 4. KRAS and RAS Family of Proteins

The transforming potential of RAS proteins has been described since their discovery in the late 1970s. The RAS family comprises 39 proteins [81]. RAS proteins belong to small GTPases family of ~21 kDa, which under physiological conditions cycle between active (GTP-bound) and inactive (GDP-bound) conformations in response to a stimulation of cell surface receptors, such as EGFR, CMET or HER family [82]. Alterations impair GTP hydrolysis, which results in a persistent activation of RAS proteins [83]. Overall, according to the COSMIC database, the prevalence of *KRAS*, *HRAS* and *NRAS* mutations is 23%, 3.41% and 7.51%, respectively, and varies in different cancer types [83,84]. Typical for pancreatic, colorectal, and lung cancer are *KRAS* alterations, whereas *HRAS* is more specific to melanoma and head and neck cancer, followed by *NRAS* in hematological malignancies [84]. 

The interest in RAS proteins research is due to the ability of the mutation-activated RAS proteins to control multiple intracellular oncogenic pathways affecting metabolism, angiogenesis, proliferation, differentiation, migration and survival. Signaling cascades are triggered by binding of RAS-GTP to RAS-binding domains within several known RAS effector pathways (Figure 3 and Figure 4), such as: PI3K-AKT-mTOR, RAF-MEK-ERK (MAP kinase cascade), TKB1-NF-κB, PLC-PKC, TIAM and RAL [85]. Liu et al. found that *KRAS* mutations, acting through the RAF-MEK-ERK pathway, upregulate the telomerase activity [86]. An upregulation of oncogenic miR-21 by mutant KRAS through activation of the RAF-MEK-ERK and PI3K-AKT pathways in the early stages of lung tumorigenesis in cell lines and mice as well as in tumors from patients, was found [87]. To promote cancer invasion, metastasis and immune evasion in pancreatic cancer, oncogenic KRAS cooperates indirectly with mutant p53 to induce overexpression of ARF6 and its effector AMAP1, involved in regulation of integrins and E-cadherin [47]. The chemoresistance observed in lung cancer patients harboring *KRAS* oncogenic mutations was suggested to be the consequence of KRAS mediated upregulation of NRF2 transcription factor, an oncogenic hub of oxidative stress response [88]. Analysis of colon cancer cell lines revealed that KRAS G12V induces *HIF1A* hypoxia sensor transcription and, in turn, overexpressed *HIF1A* or hypoxia activates KRAS signaling [89]. 

Among studies investigating the cooperation of KRAS and other factors in driving cancer initiation and progression the interplay between KRAS and CMYC has drawn attention. In 1983 Weinberg et al. observed that introducing activated RAS or MYC to rat embryo fibroblasts had no effect, only when RAS and MYC were combined were normal cells transformed and tumorigenic effect maintained in mice and rats [90]. These findings however, were not reproduced in human fibroblasts, where the activation of KRAS alone triggered senescence, whereas the activation of CMYC or KRAS and CMYC promoted apoptosis, which could not be prevented even by a loss of p53, suggesting the necessity for further changes for the full transformation [7]. Nevertheless, Kortlevel et al. exploited a mouse model to confirm that co-activation of both oncogenes led to a transition into aggressive adenocarcinomas [91]. Mechanistically, KRAS was found to maintain CMYC protein stability through an ERK1/2-related mechanism [92]. 

For many years KRAS has been considered undruggable [84]. Due to the high affinity of KRAS for GTP and a high GTP abundance in cell cytoplasm the efforts to design an inhibitor binding to KRAS competing with GTP have remained elusive. However, two recent independent studies showed that covalent inhibitors of KRAS with G12C mutations led to regression of lung and colon tumors in clinical trials [93,94]. Nevertheless, the most reliable targeted therapies against RAS-dependent cancers at the present are directed at PI3K (alpelisib), MEK (trametinib, cobimetinib and selumetinib), mTOR (everolimus) or at multiple signaling kinases (sorafenib), but not at RAS itself [95]. Thus, the oncogenic potential of RAS being defined by its multiple oncogenic pathway crossovers is reflected in the most effective strategies against the RAS-driven neoplasias.

## 5. PI3K and BRAF

Two of the KRAS downstream signal transducers—PI3 kinase (PIK3CA) and BRAF can be particularly frequently activated independently of RAS. According to the data from the TCGA database *PIK3CA* is the second most commonly mutated gene in human neoplasias, with the highest frequency in uterine, breast and head and neck carcinomas [1]. Upstream PI3K activation is dependent on binding of a ligand to receptor tyrosine kinases and further interactions with other receptors or KRAS [96]. Hot-spot mutations in *PIK3CA* were shown to induce gain-of-function, strongly activating the AKT-mTOR pathway independent of upstream oncogenic signaling (as mentioned for KRAS in the previous section) and contributing to cell transformation [97]. Likewise, the RAF kinase predominantly altered in cancer, BRAF, can act independently from upstream RAS signaling [98]. The most prevalent class I *BRAF* mutations, V600D/E/K/R, frequent especially in melanoma and thyroid or colorectal cancers enable BRAF functioning as RAS-independent monomers [99]. Class II *BRAF* mutations are localized in the P-loop or activation domain and allow signaling as RAS-independent dimers, whereas class III *BRAF* mutants depend on RAS signaling. As is the case with PI3K, *BRAF* mutation-mediated activation allows a significant proportion of the RAS downstream signaling to be activated, sufficient for supporting the growth of specific cancer types and being an attractive potential drug target. 

One isoform-specific PI3K inhibitor, idelalisib, is approved for leukemia and lymphoma treatment. Other inhibitors are being investigated in clinical trials, however due to a limited activity of single-agent therapies combination therapies have been tested, e.g., with mTOR inhibitors [100]. Despite the available mutant-specific BRAF inhibitors, such as vemurafenib and dabrafenib for V600, the crucial problem for BRAF inhibitors is resistance, and yet again combination therapies are implemented, e.g., targeting BRAF and MEK [101]. In both cases, the tested combinations targeting additional proteins vertically in the same pathway appear to only be temporary therapeutic solutions, resulting in adaptive and acquired resistance due to bypassing the targeted pathway [100,101]. Therefore, an emerging solution is to target the interplay with more separated oncogenic pathways (Section 15).

## 6. Telomerase

The human *hTERT* gene and its active product, telomerase ribonucleoprotein (protein TERT and the RNA component TERC), is estimated to be activated in 90% of human cancers [102,103] and its activation is a key oncogenic event leading to transformation [4]. However, telomerase by itself is not sufficient to actively drive cell transformation [104]. Presence of the active telomerase in cancer cells is an aberrant representation of a physiological process of counteracting chromosome telomere shortening (acting as a molecular division limiting indicator) in cells which require an unlimited replicative potential [105]. The main crossover with other driver oncogenes occurs when they become activated in the presence of the active telomerase, allowing RAS, MYC and other oncoproteins to drive other neoplasia hallmarks [4,10]. As mentioned in the MYC section, the CMYC transcriptional complex is known to activate *hTERT* transcription [106] and *CMYC* overexpression was found to immortalize epithelial cells via upregulation of *hTERT* [107]. Other pro-oncogenic transcription factors found to induce *hTERT* expression are FOXM1 [108] and ETS2 [109], whereas SYMD3 methyltransferse is facilitating this induction via histone methylation at the *hTERT* promoter [110]. The *hTERT* promoter is often a subject to activatory mutations in neoplasia which facilitate its activation by GABPA and other ETS family transcription factors. Mutation occurrence is associated with the presence of other oncogenic drivers, such as mutant BRAF or FGFR3 [111]. Interestingly, telomerase was suggested to possess extra-telomeric oncogenic activities [112] which include acting as the co-factor in the β-catenin transcriptional complex [113], anti-apoptotic activity in the mitochondria [114] or activator of the CMYC pathway [66]. These activities are possibly responsible for the limited ability of telomerase to drive cancer hallmarks apart from its role in cell immortalization.

Even though telomerase is an extremely universal oncogenic driver, attempts to exploit this fact clinically have not yielded spectacular effects. Small molecule inhibitors and vaccine-based immunotherapies which have reached clinical trials produce, at best, partial responses [115,116]. These can be potentially boosted when combined with other drugs, including checkpoint inhibitors in immunotherapy. The anti-telomerase therapy may suffer from the fact that partial or temporary telomerase targeting may not be sufficient to kill cells stimulated by more effective major oncogenic drivers, which may even bypass telomerase absence via alternative lengthening of telomeres (ALT) mechanisms [117].

## 7. Cellular Proteasome Machinery

The proteasome is the core part of the ubiquitin-proteasome system responsible for protein degradation in eukaryotic cells. An increased proteasome activity was observed in breast [118], colon [119], head and neck [120] cancers and other neoplasias. Proteasome inhibitors bortezomib and carfilzomib are approved for multiple myeloma and mantle cell lymphoma treatments [121]. The primary role of the proteasome machinery in neoplasia is postulated to be the increased degradation of tumor suppressor proteins, such as p21, p27, p53, PUMA, NOXA or IκB inhibitor of the NF-κB pathway [122] while maintaining a general protein homeostasis and oxidized protein cleanup in rapidly proliferating cells [123]. Co-chaperone CHIP links the ubiquitin-proteasome system to chaperones HSP70/HSP90, which are also responsible for proteostasis in cancer [124]. 

Genes encoding proteasome subunits are regulated primarily by NRF1 (NFE2L1) and NRF2 (NFE2L2) transcription factors [125,126], and additionally by STAT3 and NFY [127,128]. This mechanism does not only rely upon the oncogenic roles of the mentioned transcription factors but can also can be intercepted by other oncoproteins, such as mutant p53 which selectively regulates NRF2 transcriptional program, activating the proteasome expression and influencing the cellular miRNA maintenance via miRNA maturation factors [38,39]. Mutant HRAS was found to increase the level of 26S proteasome possibly as a result of the unfolded protein response (UPR) and JNK pathway activation [129]. 

26S proteasome is a multiprotein complex composed of 20S core and 19S regulatory caps. Proteins, including tumor suppressors, can be degraded in a ubiquitin-independent manner by the 20S proteasome core alone [130]. Furthermore, specific 20S core subunits may be replaced by alternative variants in the immunoproteasome for antigen peptide processing (with emerging, but yet unclear specific roles in neoplasia), whereas regulatory particle 19S can be substituted by 11S proteasome activators PA28αβ (11Sαβ) or PA28γ (11Sγ, a hexamer of REGγ/PSME3 proteins) [122]. PA28γ is overexpressed in multiple cancers and is associated with different oncogenic pathways [131]. Its expression can be induced by mutant p53 [132]. PA28γ promotes YAP and NF-κB oncogenic signaling, among others, and reinforces cross-talks between inflammation and growth pathways [133]. Activity of the WNT/β-catenin signaling pathway is enhanced by increased degradation of GSK-3β mediated by PA28γ [134]. Overexpression of *PSME3* in cancers influences the stability of CMYC, however the published results are contradictory: high levels of PSME3 inhibited CMYC degradation in pancreatic cancer cells [135], whereas studies performed in cell lines showed that PA28γ promotes CMYC degradation [136]. 

Although the described research indicates that most solid tumor-derived cells are addicted to the increased proteasome activity [129], tumors proved to be resistant to proteasome inhibitors in clinical trials [137]. Multiple myeloma patients often experience acquired resistance to bortezomib [138]. Hence, a combination of targeted therapies seems to be possible method, such as inhibiting the known compensatory mechanisms to proteasome inhibition, such as activation of HSP70/HSP90 molecular chaperones or autophagy [139,140,141], or targeting the NRF2-mutant p53-miRNA axis to counteract the proteasome inhibitor resistance [38,39].

## 8. HSP Molecular Chaperones

Another protein group, engaged in a wide range of interactions with cellular oncogenic systems is the molecular chaperones. As opposed to the proteasome machinery—which degrades accumulating proteins in a stressed environment of a cancer cell—the molecular chaperones tend to directly support the mechanisms needed for the survival and dissemination of neoplasias. The main eukaryotic molecular chaperone machinery consists of the heat shock protein HSP90 and HSP70 chaperones plus a variety of co-chaperones, responsible for folding, stabilization and a controlled degradation of proteins [142]. This cellular chaperone machinery, recently often collectively called “the chaperone”, can be reprogrammed to specifically support cancer cells, e.g., by the oncogenic transcription factor CMYC [143] and the main stress-induced chaperone gene transcription factor HSF1 [144]. HSP90 family chaperones were most often discussed in the context of cancer as some of HSP90’s known substrates are oncogenic protein kinases such as SRC, FAK, MET, EGFR, HER2 [145,146,147] as well as the telomerase [148] and the other main oncogenic drivers. These are supported either indirectly, as KRAS in the context of lung cancer [149], or as direct client proteins of the HSP chaperones, as CMYC in lymphomas [150,151] and p53 in numerous neoplasias. Interestingly, the HSP chaperone stabilization effect on mutant p53 is most likely a side effect of stalling the chaperone machinery normally supporting WT p53 on the intrinsically unstable missense p53 mutants, as described earlier in the section about *TP53* mutants [22,24,25]. 

HSP90 family proteins were the first chaperones strongly considered as targets for antitumor drugs. However, despite over a decade of phase I-III clinical trials none of the second- or third-generation HSP90 inhibitors have yet been introduced as a standard therapy [142,152]. This prompts an increased interest in targeting networks responsible for HSP90 regulation and interaction with other chaperones or oncogenes—including HSP70 which participates in most of HSP90 activities [153], HSF-1 transcription factor [154] and selected co-chaperones, such as small HSPs and HSP40 family proteins [155,156], or attempts to target HOP—an adaptor protein, linking the HSP90–HSP70 chaperone systems [143,157]. The wide compensatory regulation and substrate redundancy of molecular chaperones is likely responsible for hindering the anti-chaperone therapies in clinics and this therapeutic protocol seems to be in need of a more holistic approach rather than focusing solely on HSP90 inhibitors. 

## 9. NF-κB

Nuclear factor kappa-light-chain-enhancer of activated B cells (NF-κB) is a family of transcription factors that includes five genes encoding: NF-κB1, NF-κB2, RELA (p65), RELB and CRER. The NF-κB pathway is constitutively active in most cancers and is involved in almost all processes crucial for oncogenesis: transformation, inflammation, proliferation, avoidance of apoptosis, invasiveness, metastasis, chemo- and radio-resistance [158]. As mentioned above, the proteasome is indispensable for the NF-κB activation [159]. The NF-κB1 and NF-κB2 proteins are synthesized as large precursors, p105, and p100, which undergo processing by the proteasome to generate mature NF-κB subunits, p50 and p52, respectively [160]. NF-κB is present in the cytoplasm in an inactive form, bound by an inhibitor, IκB. Stimulation of different signaling pathways leads to activation of IKK kinase and phosphorylation of IκB, followed by ubiquitination and proteasomal degradation. Active NF-κB is translocated to the nucleus where it binds promoters of target genes. Multiple oncogenic signaling including EGFR, CKIT, HER2 receptors via RAS contribute to constitutive activation of NF-κB in cancers [158]. While KRAS activates NF-κB, NF-κB strongly contributes to the oncogenic effects of KRAS in lung cancer [161]. Importantly, deletion or mutation of p53 is required for the RAS-dependent NF-κB activation [161,162]. The GOF p53 mutants prolong the activity of NF-κB induced by TNF-α contributing to the onset of inflammation-associated colon cancer [163]. In B cell lymphomas a cross-reaction of CMYC and the constitutively active NF-κB leads to a more aggressive disease than the overactive NF-κB alone [164]. In HER2-positive breast cancer, upon irradiation, NF-κB activates CMYC transcription which in turn binds to hTERT promoter [165]. 

NF-κB is the oncogenic driver whose indirect inhibition dominated therapeutic attempts, as direct targeting is known to cause adverse effects due to housekeeping and tumor suppressive roles of NF-κB [158]. Indirect targeting is an example of taking advantage of the known interplays of NF-κB with other oncogenic factors, which includes approved drugs inhibiting mechanisms mentioned above, e.g., proteasome, HER2 and EGFR.

## 10. AP-1—FOS/JUN

Activator Protein 1 (AP-1) leucin-zipper dimeric transcription factors were discovered in the late 1980s as possible promoters of cell proliferation and transformation [166]. AP-1 proteins are primarily composed of different combinations of JUN and FOS protein families’ members, but also include ATF and MAF family proteins [167]. Even though for the most of the 1990s AP-1, especially in its CFOS/CJUN setup, was considered an oncogenic transcription factor in vitro and in vivo [168,169], selected variants of AP-1, dependent on the subunit composition, are not oncogenic or may possess tumor suppressive activities [170]. Thus, the question has arisen as to which molecular events push AP-1 onto the oncogenic track. Earliest evidence pointed to a direct activation of oncogenic AP-1 variants by RAS family proteins [171,172] and their downstream signal transducers phosphorylating AP-1 proteins—MEK1 [173] and JNKs [174]. The list of oncogenic crossovers of AP-1 soon became longer, with NF-κB pathway cooperation [175] or activation by and cooperation with the GOF p53 mutant variants [176]. The downstream AP-1 oncogenic targets include invasiveness-related MMP genes [177], proliferation-stimulating EGF pathway genes [178] and oncogenic miRNAs such as miR-21 [179]. More recent findings, involving genomics and transcriptomics point to the AP-1 dimers as regulators of enhancer-associated rather than promoter-related global transcription reprogramming. In these studies, the AP-1 has been found to cooperate with oncogenic transcription factors YAP/TAZ [180] and ETS family proteins [181]. 

The ambiguous nature of AP-1 determined by a delicate balance in the subunit composition is most likely to blame for a decline in the number of studies since 1990s considering AP-1 as a promising cancer-specific therapeutic target and diagnostic factor. However, recent reports show effectiveness of an AP-1 small molecule inhibitors, often designed years earlier in cancer preclinical models, e.g., in head and neck or lung cancers [182,183], which may be yet included on maps of specific or combinational anti-cancer therapies. 

## 11. β-catenin and the WNT Pathway

The wingless-type MMTV integration site family (WNT) pathway, discovered as a key developmental signaling pathway, entered the cancer research scene in early 1990s with finding that *APC* (adenomatous polyposis coli) tumor suppressor gene mutations contribute to activation of WNT in colon cancer [184]. The pathway’s tumorigenic effects are fueled primarily by the transcription co-factor—β-catenin. The pathway activation was found in many neoplasias outside colorectal cancer—such as leukemia [185], melanoma [186] and breast cancer [187]. The number of links of both β-catenin-dependent and other, so called non-canonical WNT signaling routes, and crossovers with other oncogenic signaling grew, demonstrating that WNT is one of the most entangled pathways known in molecular oncology [188]. 

Apart from the inactivation of APC and other negative regulators, β-catenin can be hyperactivated in the context of neoplasia by multiple oncogenic signals: by the canonical membrane receptor Frizzeled (WNT ligands), by LGR receptors (R-spondin ligands) [189], by EGFR via the SRC/PKM2 axis [190] and by the prostaglandin 2 receptor EG2 [191]. When released from the cytoplasmic destruction complex, β-catenin translocates to the nucleus and, not possessing its own DNA biding domain, cooperates with partners, primarily with lymphoid enhancer factor (LEF) and T-cell factor (TCF), recruiting epigenetic modifiers such as CBP/p300, BRG1 and BCL9 [184,192]. However, this process may also involve stimulatory members of other oncogenic pathways - PCNA-associated factor (PAF) [193] or YAP [194]—the latter also involved in a suppression of β-catenin in the cytoplasm (see the YAP/TAZ paragraph). β-catenin’s transcriptional activity was also found to be activated by mutant p53 [195] and KRAS [196]—although these results need further assessment of specificity, as the mutant p53 and KRAS axes were shown to be independent or to antagonize β-catenin [197,198]. The transcriptional complex controlled by β-catenin is known to induce multiple oncogenic drivers and modulators mentioned in other sections of this review, including CMYC [199], CJUN [200] and hTERT [201], of which the latter was suggested to work in a telomere-independent manner with β-catenin in the nucleus [113]. The interface of the WNT pathway with other oncogenic pathways is widened by the non-canonical WNT signaling, independent of β-catenin [188]. These signaling routes involve JNK-mediated AP-1 activation [202,203] or NFAT transcription factor induction by calcium and a Calcineurin (CaN)-mediated mechanism [204,205] However, the WNT non-canonical signaling routes were mainly described outside of the neoplasia context, requiring further assessment as drivers of neoplastic phenotypes [188]. 

The involvement of the WNT pathway in normal tissue or developmental processes makes it a difficult therapeutic target [206]. Hence, while inhibitors of β-catenin interactions with TCF and CBP were tested mostly in cancer preclinical models [207,208], they reach clinical trials often combined with chemotherapeutics or inhibitors of other oncogenic pathways [209]. Again, for the WNT pathway, the oncogenic cooperation proves to be a decisive factor for its clinical testing options in neoplasia. 

## 12. YAP/TAZ

Yes-associated protein (YAP) and PDZ-binding motif protein (TAZ) are transcription factors which play a key role in a regulation of cell mechanotransduction, organ size, tissue growth and regeneration, and consequently, in tumorigenesis. Their physiological function has become understood relatively recently, as functional homologues of a signal transducer Yki of the Hippo pathway in *Drosophila melanogaster* [210]. Many reports demonstrated an increased nuclear localization and activation of YAP/TAZ in cancers of different organs including the liver, breast, lung, pancreas and skin [211]. Among YAP transcriptional targets in the context of cell transformation several oncogenic pathway proteins have been identified, such as inhibitors of apoptosis BIRC5 and BCL2L1 [194], amphiregulin (AREG)—an epidermal growth factor (EGF) family member [212] and connective tissue growth factor (CTGF) [213]. The Hippo pathway is conserved in mammals and multiple negative regulators of YAP/TAZ in this pathway are known tumor suppressors in humans, including Merlin (NF2) protein and LATS kinases which cause cytoplasmic retention of YAP/TAZ [214]. However, YAP/TAZ activation in cancer is not dependent exclusively on the inactivation of the Hippo pathway suppressors. Mechanical cues can be transduced to YAP/TAZ independent of the Hippo pathway, via a Rho GTPase activity and actomyosin cytoskeleton stiffness [215]. YAP was found to be activated both independent and dependent on the Hippo pathway by mutated *GNAQ* and *GNA11* oncogenes respectively, resulting in a growth stimulation of uveal melanoma cells [216,217]. In breast cancer cells, YAP/TAZ were found to be under the control of the steroid synthesis mevalonate pathway that is often aberrantly upregulated in neoplasia [218]. In pancreatic cancer, both the activation of YAP by mutant KRAS and the redundancy of YAP and mutant KRAS pathways have been discovered in a KRAS-driven mouse neoplasia model [219]. Eventually, other key oncogenic transcription factors—CMYC and mutant p53—were found to cooperate with YAP/TAZ [32,220], along with AP-1/TEAD and β-catenin mentioned in earlier sections. The picture is further complicated by YAP and TAZ being found to possess tumor suppressive activities in some cancer contexts—by suppressing the WNT activity or triggering stress-induced apoptosis [221]. Thus, therapeutic attempts to target oncogenic YAP/TAZ activities are in the early stages. They include in vitro and in vivo preclinical tests of YAP-TEAD complex inhibitors verteporfin [216,217] and Super-TDU [222], but also the interplay with other pathways, such as targeting of a YAP-BRD4 complex with BET inhibitors [223], and a use of statins as inhibitors of YAP/TAZ-activating mevalonate pathway [218]. This latter protocol potentially has the highest clinical significance, as statin treatment is generally regarded as protective against neoplasia. This also means, however, that specificity of anti-YAP/TAZ targeted therapies are far from being clinically understood and exploited.

## 13. Non-coding RNAs (Micro RNAs and Long Non-codong RNAs)

The advances from the non-coding RNA (ncRNAs) research field over the last decades led to discovery of the function of selected ncRNAs as oncogenic drivers—including 19-24 nucleotide micro RNAs (oncomiRs) and long non-coding RNAs (lncRNAs) with a length of over 200 nucleotides. Whereas the ncRNAs have been increasingly used as diagnostic markers, researchers are still trying to determine whether and which of the ncRNAs can be regarded as actual oncogenic drivers, decisive for cancer hallmark phenotypes [224]. We here list the candidates that work through crossovers with the other driver pathways.

miR-21, one of the first discovered mammalian miRNAs regarded as one of the most universal oncomiRs, is overexpressed in various types of neoplasia [225]. It is mostly known to act by interfering with the expression of tumor suppressors which causes the activation of oncogenic pathways. For example—targeting PTEN expression, allows for miR-21-mediated activation of the PI3K-AKT pathway [226]. miR-21 also enhances oncogenic KRAS activity and tumorigenesis by targeting negative KRAS pathway regulators in non-small cell lung cancer [227]. Similar are the cases of other universal oncomiRs—they mostly support oncogenic pathways by either repression of translation or promotion of mRNA degradation of oncosuppressors, e.g., miR-31 downregulates the WNT pathway antagonists and contributes to breast tumorigenesis [228], miR-221 targets DDIT4 transcript, a negative regulator of the mTOR pathway, contributing to liver tumorigenesis [229], miR-155 inhibits C/EBPβ, leading to *HK2* overexpression and the metabolic support of cancer cells [230]. OncomiRs expression was found to be under the control of multiple oncogenic drivers. Mutant p53 affects miRNA expression both at the level of maturation (mentioned in Section 2) and the direct transcription control, including miR-155 [231], miR-128-2 [232] or broader miRNA signatures, which include miR-205 and miR-21 [233]. CMYC, apart from being controlled by multiple ncRNAs (see below) activates miRNA transcription of important oncogenic miRNA clusters in cooperation with E2F family transcription factors [234]. RAS family gene pathways, while sending multiple activatory signals to transcription factors activating oncomiRs, including CMYC, are also extensively regulated by a complex miRNA network, which mostly of tumor suppressive in nature, and is downregulated in neoplasia [235]. 

lncRNAs play a significant role in the regulation of gene expression at the transcriptional and post-transcriptional level by targeting, for example, p53, NF-κB, SOX2 and OCT4 [236]. H19, one of the first identified lncRNAs in the early 1990s, was found to be involved in a tumor development promotion, being activated by oncogenic pathways such as PI3K/AKT [237], acting by sponging notable tumor suppressive miRNAs, for example let-7 [238]. LncRNA MALAT1 (NEAT2) sequesters miR-103-3p and upregulates expression of the MCL1 oncoprotein in leukemic cells [239]. Mutant p53 cooperates in a splicing regulatory complex with ID4 protein, SRSF1 splicing factor and MALAT1 to promote basal-like breast cancer [240]. MALAT1 is also implicated in upregulating mutant KRAS expression by sponging miR-217 in pancreatic adenocarcinoma [241]. A similar mechanism for mutant KRAS pathway activation was found for other lncRNAs in pancreatic cancer—NUTF2P3-001 [242] and UCA1 [243]. Oncogenic lncRNA PVT1 was found to increase expression of ATG7 and Beclin1 by targeting tumor suppressive miR-186, promoting glioma [244]. PVT1 is also involved in the regulation of CMYC by supporting its stability via a possible direct interaction, to augment cell proliferation [245]. Other oncogenic lncRNA EPIC1 was shown to indeed directly interact with the CMYC protein and induce its transcriptional activity [246]. HOTAIR is yet another lncRNA implicated in CMYC activation [247], and is also engaged in broader cancer-promoting chromatin state reprogramming [248], which is the mechanism known to involve other lncRNAs such as FAL1 [249]. CCAT2, the lncRNA that promotes colorectal cancer growth, metastasis, and chromosomal instability upregulates WNT and CMYC signaling when activated by the WNT pathway [250], but also by the KRAS-MEK axis in pancreatic cancer [251]. CMYC is involved in the transcriptional activation of oncogenic lncRNAs, including PVT1 [245], CCAT1 and other related non-coding loci [252]. Mutant p53 in conjunction with the YAP/TEAD transcription-competent complex is involved in an upregulation of a circular form of oncogenic PVT1 [253]. The currently known interrelations of mutant p53 with several oncogenic ncRNAs are representative examples of a still limited but rapidly expanding knowledge of the interplay of the main driver oncoproteins with ncRNAs [254]. This knowledge will surely grow significantly in the nearby future as over 30 thousand lncRNAs alone are found in a typical human transcriptome.

Due to advanced vectors and the unstable molecules needed to inhibit oncogenic ncRNAs, successful attempts to use antisense RNA oligonucleotides (ASOs) are so far mostly preclinical, despite a rising number of clinical trials [224]. Such is the case of antiMirs/antagoMirs targeting miRNAs described above [255,256] or antisense nucleotides against lncRNA MALAT1 [257]. Noticing that broader mechanisms of ncRNA deregulation in cancer are crossing over to other targetable oncogenic pathways could help in making the ncRNA targeting more robust. For example, the mentioned CMYC or mutant p53 effects on the activity and maturation of wide miRNA populations [252,254] can be theoretically co-targeted along with selected oncomiRs. 

## 14. Exploitation of the Oncogene Cooperation in Clinics

The biggest challenge resulting from the accumulating knowledge on the networks of interaction between the driver oncogenes is putting it to a diagnostic and therapeutic use. Until now standard clinical protocols that include simultaneous targeting of more than one oncogenic pathway have been lacking. Clinically approved oncogene combination targeting therapies usually involve actions against components of the same pathway vertically—BRAF and MAPK/ERK kinase (MEK) inhibitors [258]—or inhibiting multiple targets activating uniform signaling routes. This approach includes double antibody treatments aimed at inhibiting oncogenic receptors [259], or attempts to overcome resistance to anti-EGFR therapies with sorafenib—a multi-kinase inhibitor of BRAF, PDGFRβ, and VEGFR2, which are all connected primarily with the KRAS-BRAF/PI3K signaling [260,261].

Most of the genuine strategies that target more than one pathway are preclinical—oncogene network-specific examples and ideas were briefly described in each of the previous paragraphs. Many of them however do not produce positive responses in clinical trials. The example is the case of targeting oncogenic pathways in combination with proteasome inhibitors, attempting to reposition them from the multiple myeloma treatment [262], such as the poor result of the pancreatic cancer-focused trial of proteasome inhibitor marizomib with inhibitors of HDACs), which represent another, heterogeneous group of oncogenic drivers—the epigenetic modulators [263]. 

However, encouraging cases have been reported. The development of resistance for both BRAF and BRAF-MEK inhibitor therapies provoked a search for additional druggable interplays with oncogenic pathways resulting in finding the molecular chaperone machinery. This resulted in preclinical tests of simultaneous BRAF and HSP90 inhibition [264] and later a successful phase I clinical trial involving unresectable BRAF mutant melanoma [265]. This case showed that a rational approach to understanding inter-oncogene connections can lead to rapid drug repositioning. If the map of oncogenic interactions (Figure 4) is used together with neoplastic tissue-related treatments—such as immunotherapies or tumor angiogenesis-modulating approaches—their usefulness may further increase. Examples of this strategy include the combination of BRAF/MEK inhibitors with immune checkpoint inhibitors (promising clinical trials are under way [266]) or with vascular normalization-promoting VEGF inhibitors [267]. 

The field of transformed cell competition is still largely unexplored, where the spatial and functional requirements of a given cell population are driven by a particular oncogenic program which allows them to dominate the neoplastic tissue [268] or leads to intra-tumoral heterogeneity [269]. This suggests that also within cells, selected oncogenic programs may compete rather than cooperate to overtake driving at different transformation stages or in cell subpopulations—which is a potentially important, yet uncharted, therapeutic choice indicator. 

Table 1 shows upstream and downstream interplays of the universal oncogenic drivers described in the text along with the current and proposed therapeutic solutions.

## 15. Conclusions

This review does not address the details of mechanistic features of the oncogenic drivers, but rather focuses on their interplays. As described above many of these interplays are in fact basic and essential modes of function of the major pro-neoplastic genes and proteins. The phenotypic effect of the driver oncogene is largely defined by its network of partners, compensators as well as competitors – the latter of which include tumor suppressors and possibly other oncogenes. 

Even though the anti-neoplasia therapeutic landscape is currently growing far beyond targeting the oncogene-related cell-autonomous processes—better understanding of the intracellular driver oncogene interplay and filling the remaining gaps in the map of their molecular connections (Figure 1, Figure 2, Figure 3 and Figure 4) will help to gradually bring more efficient and personalized therapies to the patients. 

## Figures and Tables

**Figure 1 cancers-12-01532-f001:**
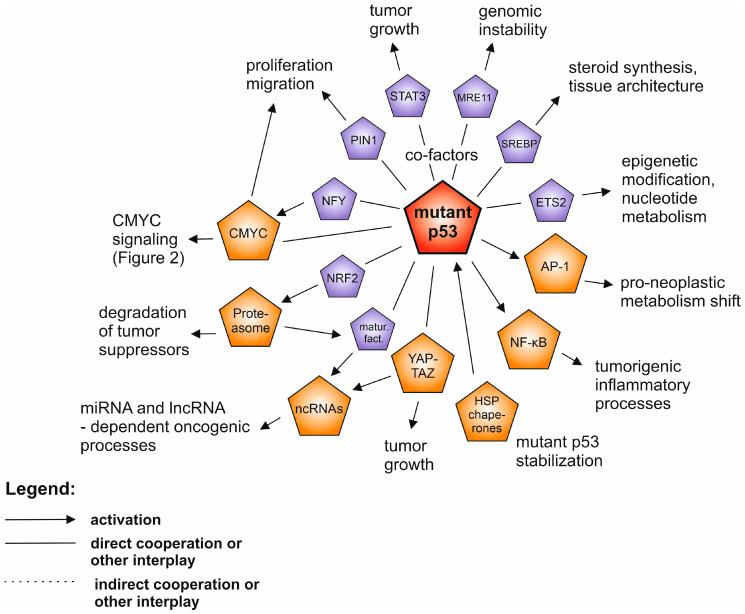
The network of the interplay between mutant p53 (highlighted in red) and other oncogenic drivers in human neoplasia. The drivers described in separate paragraphs are shown in orange, while additional oncoproteins are shown in purple. Only the selected and well supported connections are shown, as described in the text.

**Figure 2 cancers-12-01532-f002:**
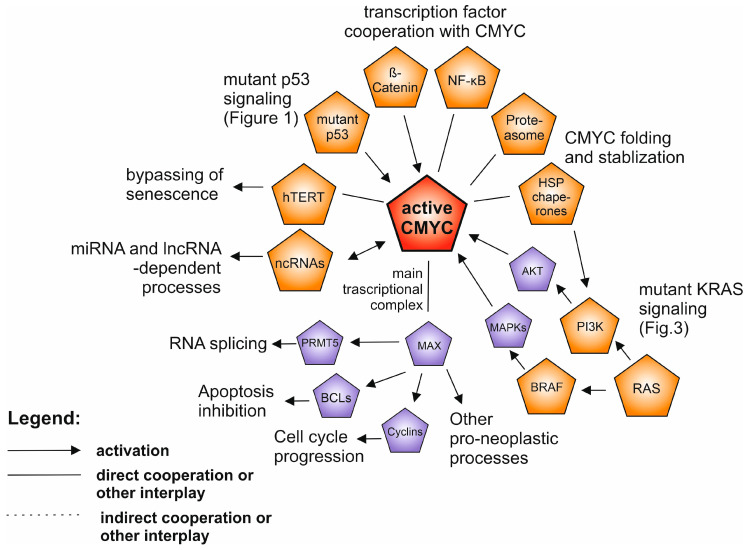
The network of the interplay between activated CMYC (highlighted in red) and other oncogenic drivers in human neoplasia. The drivers described in separate paragraphs are shown in orange, while additional oncoproteins are shown in purple. Only the selected, well supported connections are shown, described in the text.

**Figure 3 cancers-12-01532-f003:**
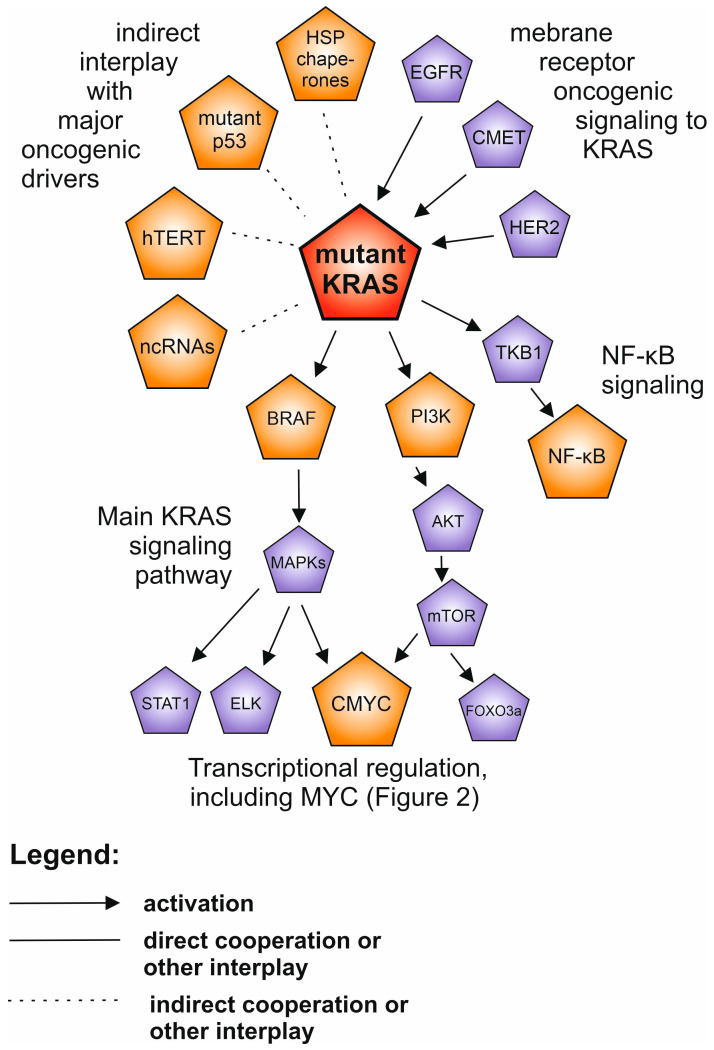
The network of the interplay between mutant KRAS (highlighted in red) and other oncogenic drivers in human neoplasia. The drivers described in separate paragraphs are shown in orange, while additional oncoproteins are shown in purple. Only the selected and well supported connections are shown, described in the text.

**Figure 4 cancers-12-01532-f004:**
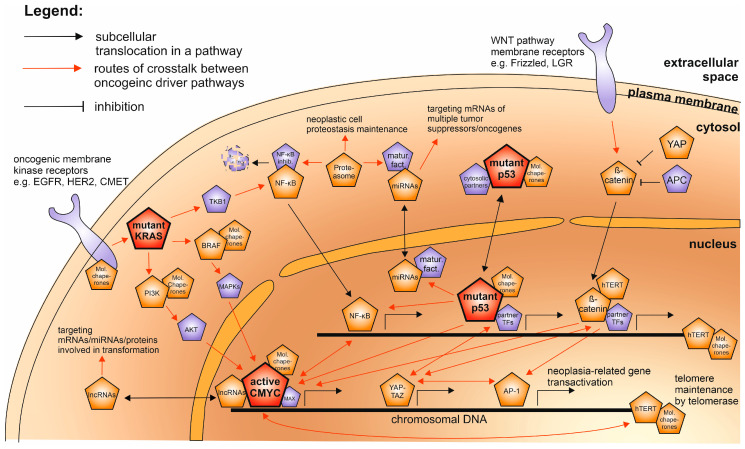
The subcellular localization of the interplay network of the human oncogenic drivers as described in the text. The most decisive and universal drivers—mutant p53, KRAS, and hyperactive CMYC—are highlighted in red; the rest of the universal drivers are in orange and additional oncogenic partner factors described in the text are shown in purple. The links between oncoproteins of different pathways are shown either by red arrows or, when a direct protein complex is formed, by overlapping symbols.

**Table 1 cancers-12-01532-t001:** Upstream and downstream interplays of the universal oncogenic drivers described in the text along with the current and proposed therapeutic solutions.

Oncogenic Driver Protein Name	Frequency/Mode of Activation in Multi-Neoplasia Studies	Upstream and Partner Oncogenic Factors Crosstalk	Downstream Oncogenic Process/Pathway/Target Examples	Direct Clinical Options in Neoplasia	Examples of Proposed/Tested Combinations with Other Targeted Therapies	Refs.
Mutant p53	42–43% mutated (of which >70% are missense—potential gain of function driver mutations)	Molecular chaperones, transcription co-factors: ETS2, SREBP1/2, NRF2, NFY, NF-κB, STAT3, YAP/TAZ; other partners: MRE11, Drosha/Dicer, DAB2IP	Genomic stability, steroid synthesis, epigenetic alterations, nucleotide synthesis, proteostasis, miRNA maturation, cell cycle/proliferation, migration/invasion	APR-246 (PRIMA-1 MET) in clinical trials	HSP90 inhibitors, HDAC inhibitors, proteasome inhibitors, statins	[11,12,13,14,15,16,17,18,19,20,21,22,23,24,25,26,27,28,29,30,31,32,33,34,35,36,37,38,39,40,41,42,43,44,45,46,47,48,49,50,51,52,53,130,132,162,163,176,194,196,231,232,233,240,253,254]
CMYC	5–20% mutated (of which >70% are amplifications), estimated hyperactive in majority of neoplasias	Transcription co-factor MAX, Pathways: Hedgehog, WNT, NOTCH, KRAS, PI3K, JAK-STAT3, MAPK-HNRP and mTORC1-S6K1, ncRNAs	Cell cycle (cyclins), metabolism (LDHA), apoptosis (BCL-XL), RNA splicing (PRMT5), immortalization (hTERT)	No direct inhibitors in clinics for neoplasia treatment (only preclinical).	CMYC-MAX complex inhibitors, inhibitors of RAS/PI3K/BRAF-related signaling	[7,8,9,10,29,33,34,49,55,56,57,58,59,60,61,62,63,64,65,66,67,68,69,70,71,72,73,74,75,76,77,78,79,80,91,92,106,107,109,135,136,150,151,164,165,199,220,234,245,246,247,252]
KRAS	7–17% mutated (up to 90% of which are missense driver mutations)	Receptor kinases (e.g. EGFR, CMET, HER2) (mutations in RAS provide independence)	metabolism, angiogenesis, proliferation, diffe-rentiation, migration via PI3K-AKT-mTOR, RAF-MEK-ERK, TKB1- NF-κB pathways, CMYC, hTERT	Small molecule KRAS G12C inhibitors (AMG 510, MRTX849) in clinical trials	Inhibitors of up- or downstream modulators (receptor kinases, PI3K, MEK, mTOR, BRAF etc.)	[4,5,6,7,47,48,50,64,65,71,81,82,83,84,85,86,87,88,89,90,91,92,93,94,95,162,171,172,179,181,195,197,219,226,234,241,242,243,260]
PI3 kinase	10–18% mutated (>80% of which are missense driver mutations)	Receptor tyrosine kinases, RTK, GRB2, KRAS (mutations in PIK gene provide independence), molecular chaperones	Proliferation/metabolism via AKT-mTOR pathway	Idelalisib (in standard protocol); multiple inhibitors tested in clinical trials	Downstream pathway components targeting (mTOR inhibitors), parallel pathway targeting (BRAF pathway inhibitors)	[64,85,96,97,100,173,226]
BRAF	1.5–7% (majority are missense driver mutations)	Receptor tyrosine kinases, KRAS (mutations in BRAF gene provide independence), molecular chaperones	Proliferation/metabolism via MEK-ERK pathway	BRAF mutant inhibitors in standard protocols (e.g., vemurafenib and dabrafenib)	Upstream and downstream pathway components (EGFR or MEK inhibitors), parallel pathways (PI3K inhibitors), HSP90 inhibitors, immuno-therapy, angiogenic modulators	[98,99,101,258,264,265,267]
Telomerase (hTERT)	15% mutated (>80% of which are promoter activating mutations); estimated active in 90% of neoplasias	CMYC, FoxM1, ETS family, SYMD3, molecular chaperones	Telomere-mediated cellular immortality; telomere independent activities: β-catenin/WNT, CMYC, mitochondrial apoptosis inhibition	Small molecule inhibitors (e.g., GRN163L) and peptide-based vaccines (e.g., GV1001) in clinical trials	Tested in combinations with immunotherapies and chemotherapy	[4,5,6,7,51,66,86,102,103,104,105,106,107,108,109,110,111,112,113,114,115,116,117,148,165,201]
Proteasome machinery (26S, 20S proteasome, immunoproteasome)	Mutations rare, estimated hyperactive and addictive in most neoplasias	NRF1, NRF2, STAT3, NFY, mutant p53, ubiquitin ligases and associated enzymes	Protein homeostasis (including inhibition of tumor suppressive pathways); activation of NF-κB and WNT pathways, molecular chaperones	Inhibitors in standard protocols (Bortezomib, Carfilzomib), other inhibitors tested	Autophagy inhibitors, HDACs inhibitors, oncogenic kinases inhibitors, NRF1/NRF2 inhibitors, mutant p53 targeting	[38,118,119,120,121,122,123,124,125,126,127,128,129,130,131,132,133,134,135,136,137,138,139,140,141,159,160,262,263]
Molecular chaperones (HSP90 and HSP70 families and co-chaperones)	Mutations rare, estimated addictive in most neoplasias	HSF1, HOP, small HSPs and HSP40 family co-chaperones	Client proteins including: mutant p53, kinases (e.g., SRC, CMET, EGFR, HER2), hTERT, CMYC, proteasome machinery activity	Multiple HSP90 and HSP70 inhibitors in clinical trials	HSF-1 inhibitors, co-chaperone inhibitors (e.g., targeting HOP or HSP40 proteins), oncogenic kinase inhibitors	[22,24,25,124,142,157,263,264]
NF-κB	Overexpressed in 1–12% neoplasias; activatory kinases overexpressed in 1.5–18% neoplasias	EGFR, RAS family, HER2, proteasome	CMYC, cyclins, cytokines influencing cell proliferation, survival, angiogenesis and metastasis	No direct inhibitors in clinics for neoplasia treatment (only preclinical)	Targeting of upstream pathways – proteasome, RAS/PI3K/BRAF-related pathways, receptor kinases (e.g., HER2 or EGFR)	[40,120,133,158,159,160,161,162,163,164,165,175]
AP-1 (primarily FOS-JUN family protein dimers)	Mutations in FOS and JUN genes below 1%; estimated addictive in a large proportion of neoplasias	RAS family, MEKs, MAPKs, NF-κB, mutant p53, YAP/TAZ, ETS family	Invasiveness (MMP genes), proliferation (EGF pathway genes), oncogenic miRNAs (e.g., miR-21).	No direct inhibitors in clinics for neoplasia treatment (only preclinical)	Preclinical repositioning of AP-1 inhibitors tested in other diseases (e.g., T-5224 or SR 11302), rarely combined with other therapeutics	[166,167,168,169,170,171,172,173,174,175,176,177,178,179,180,181,182,183]
β-catenin and WNT pathway	7–10% inactivating mutations in APC, <3% mutations in other WNT pathway proteins, estimated hyperactive especially in tumor-generating neoplasias	Receptor-ligand pairs: FZD-WNT, LGR/R-spondin, EGFR-EGF, EG2-PGE2; β-catenin destruction complex and nuclear co-factors: LEF, TCF, PAF, YAP1, hTERT	Cell proliferation and tumor growth via CMYC, CJUN, hTERT, targets of AP-1 or NFAT (in non-canonical signaling)	Repositioning of approved anti-WNT drugs to cancer treatment (e.g., Niclosamide, Sulindac), other inhibitors in trials	Combination with PI3K/AKT/mTOR inhibitors, multikinase inhibitors (e.g., sorafenib) and chemotherapeutics	[113,134,184,185,186,187,188,189,190,191,192,193,194,195,196,197,198,199,200,201,202,203,204,205,206,207,208,209]
YAP and TAZ	1–1.5% mutated YAP1 (up to 40% amplifications); estimated hyperactive in most tumor-generating neoplasias	Rho GTPases and cytoskeleton stiffness, GNAQ and GNA11, mevalonate pathway, β-catenin, CMYC, mutant p53, BRD4, AP-1, TEADs	Inhibition of apoptosis (e.g., BIRC5 and BCL2L1), tissue growth/stiffness/cell proliferation (e.g., AREG, CTGF)	No direct inhibitors in clinics for neoplasia treatment (only preclinical)	BET inhibitors, statins	[32,133,180,210,211,212,213,214,215,216,217,218,219,220,221,222,223]
non-coding RNAs (micro RNAs and long non-coding RNAs)	Copy number alterations, SNPs and promoter mutations so far found in single studies	miRNA maturation factors (Dicer, Drosha, DGCR8, KSRP) and their regulators e.g., mutant p53, proteasome; upregulators of specific ncRNA transcription or stability—e.g., CMYC	Cell proliferation (e.g. via RAS and PI3K-AKT pathway, CMYC, MCL1), tumor growth (e.g., via WNT pathway), cancer cell metabolic support (e.g., via STAT1, ATG7, Beclin1), chromatin reprogramming	Inhibition of oncogenic miRNAs (e.g., miR-155 by MRG-106) or targeting cells with overexpression of H19 lncRNA (by BC-819) in clinical trials	Next to none tested so far; suggested are downstream pathways and upstream regulators, such as CMYC pathway or mutant p53-proteasome-KSRP axis	[38,44,67,68,69,224,225,226,227,228,229,230,231,232,233,234,235,236,237,238,239,240,241,242,243,244,245,246,247,248,249,250,251,252,253,254,255,256,257]

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
