# Peer review of "A Driver Never Works Alone—Interplay Networks of Mutant p53, MYC, RAS, and Other Universal Oncogenic Drivers in Human Cancer"

_cancers, 2020, doi:10.3390/cancers12061532_

Round 1
Reviewer 1 Report
This review is a comprehensive and important contribution to the literature. The citations are extensive, and the thesis that multiple factors and pathways interact with oncogenes such as mutant p53 is highly relevant to understanding carcinogenesis in general.
I have a few minor recommendations. The authors discuss the effects of GOF p53 on gene transcription mainly from the perspective of its aberrant protein/protein interactions with other transcription factors that control cMyc expression. However, I suggest that the loss of the genomic repressive effects of WT p53 by GOF mutants is equally important. For example, WT p53 directly represses the transcription of cMyc and other oncogenes through binding to promoter sites on DNA. GOF p53 mutants, primarily occurring as a result of missense mutations in the DNA binding domain of WT p53, often lose the ability to bind to DNA. Hence, the enhanced expression of factors such as cMyc is due to loss of WT p53 suppressive activity as well. This should be discussed as a mechanism of carcinogenesis along with novel protein interaction functions of GOF p53. Perhaps a few sentences about the structure and function of WT p53 as a tetrameric Zn binding transcription factor should be added to the p53 section so that the consequences of the recurrent missense mutations in the DNA binding domain can be appreciated by the reader.
Author Response
Dear reviewer;
We are very grateful for appreciating our work on the synthesis of the knowledge on cooperation of the oncogenes and for providing additional suggestions to improve the manuscript. In response to your suggestions we added short text about wt p53 mechanism of action as a tumor suppressive transcription factor and about the p53 LOF mechanism of oncogene “de-suppression”.
We kept these short (sentences marked in red in the section 2. about mutant p53) as the mechanism of p53 loss of function is by definition not an active gain of oncogenic property. Profile of our review is focused on interactions of active oncogenes, while the mechanism of WT p53 LOF followed by activation of inhibited oncoproteins, such as CMYC, KRAS or telomerase (we mention those as examples in the three newly added citations – 49-51) is in fact a typical mode of a tumor suppressor inactivation - which often results in a lost repression of multiple oncogenic pathways and allows these pathways to promote neoplastic phenotypes. We are not eager to call this mechanism a bona fide gain of oncogenic function of a given protein. Thus we prefer to underline in the paragraph on mutant p53 GOF the activities of mutant p53 which indeed include an active contribution to the modulation of other oncogenic pathways.
Additionally, as suggested by another reviewer the manuscript was submitted to an English editing service to improve the grammar/syntax. Only the more important changes are marked in red, while smaller corrections such as the articles, commas, spelling etc. are simply introduced to the text with no special marking, for clarity.
With very best regards;
Dawid Walerych and the team of authors
Reviewer 2 Report
This is a well-written, high-level overview of major oncogenic pathways. The authors have done an outstanding job of synthisizing complex data from broad publications and long experimental histories. The paper is well-organized and provides helpful insight to clinicians, students, and scientists.
Author Response
Thank you very much for appreciating our work on the synthesis of the knowledge on cooperation of the oncogenes and understating the necessary omissions while writing on such a broad subject. As suggested we have submitted the text to an English editing service to improve the grammar/syntax and tried to hunt down all the small errors. Only the more important changes, mainly suggested by another reviewer, are marked in red, while smaller corrections such as the articles, commas, spelling etc. are introduced to the text with no special marking, for clarity. We indeed hope that the paper will be as useful for scientists, MDs and students, as it was for us when preparing it.
With very best regards;
Dawid Walerych and the team of authors